# Engaging community pharmacists to eliminate inadvertent doping in sports: A study of their knowledge on doping

Nontharit Voravuth[1], Eng Wee Chua[2], Tuan Mazlelaa Tuan Mahmood[1], Ming Chiang Lim[1,3], Sharifa Ezat Wan Puteh[4], Nik Shanita Safii[5], Jyh Eiin Wong[5], Ahmad Taufik Jamil[6], Jamia Azdina Jamal[2], Ahmad Fuad Shamsuddin[7]*, Adliah Mhd Ali[1]*

1 Centre for Quality Management of Medicines, Faculty of Pharmacy, Universiti Kebangsaan Malaysia, Kuala Lumpur, Malaysia, 2 Drug and Herbal Research Centre, Faculty of Pharmacy, Universiti Kebangsaan Malaysia, Kuala Lumpur, Malaysia, 3 Hospital Sultan Haji Ahmad Shah, Temerloh, Pahang, Ministry of Health, Malaysia, 4 Faculty of Medicine, Universiti Kebangsaan Malaysia, Kuala Lumpur, Malaysia, 5 Centre for Community Health Studies, Faculty of Health Sciences, Universiti Kebangsaan Malaysia, Kuala Lumpur, Malaysia, 6 Faculty of Medicine, Universiti Teknologi MARA, Sungai Buloh Campus, Sungai Buloh, Selangor, Malaysia, 7 Faculty of Pharmacy and Health Sciences, Universiti Kuala Lumpur Royal College of Medicine Perak, Ipoh, Perak, Malaysia

* adliah@ukm.edu.my (AMA); fuad.shamsuddin@unikl.edu.my (AFS)

**Data Availability Statement:** All relevant data are within the paper and its Supporting Information files.

## Abstract

This study aimed to evaluate the community pharmacists' knowledge of tackling the issue of inadvertent doping in Malaysia. A cross-sectional survey was conducted among 384 community pharmacists working in Malaysia using a self-administered questionnaire. All the respondents were pharmacists fully registered with the Pharmacy Board of Malaysia and had been working in the community setting for at least one year. Of the 426 community pharmacists approached, 384 community pharmacists participated in this study, giving a response rate of 90.14%. The majority of the respondents were females (63.5%), graduated from local universities (74.9%), with median years of practising as a community pharmacist of six years (interquartile range, IQR = 9 years). The respondents were found to have moderate levels of doping-related knowledge (median score of 52 out of 100). Anabolic steroids (95.8%), stimulants (78.6%) and growth factors (65.6%) were recognised as prohibited substances by most of the respondents. Around 65.9% did not recognise that inadvertent doping is also considered a doping violation. Most of them (90%) also have poor levels of knowledge of doping scenarios in the country. Community pharmacists in Malaysia have limited knowledge in the field of doping. More programmes and activities related to doping and drugs in sports should be held to enhance the community pharmacists' knowledge on the issue of inadvertent doping.

## Introduction

Inadvertent doping is an issue where an athlete records a positive drug test after having unintentionally and unknowingly taken a banned substance [1]. Athletes may become ill or injured,

**Funding:** This study was supported by a research grant of UNESCO Anti-Doping Fund (4500340762-A2) and Universiti Kebangsaan Malaysia (UKM-NF-2018-001).

**Competing interests:** There is no conflict of interest for all the authors.

or suffer from chronic medical conditions [2], which may necessitate the use of medications that are banned in sports. For instance, in Japan, many over-the-counter medications contain prohibited substances; this explains the high rate of unintentional doping in the country [3]. A well-known case of inadvertent doping caused by using over-the- counter medications was at the 2000 Olympic Games. Pseudoephedrine, a banned substance, was mistakenly given to a Romanian gymnast by her team physician to treat her cold symptoms. This resulted in her being stripped of the gold medal that she won, and the team physician was expelled from the Games [4]. Some successful Malaysian badminton players, weightlifters, and wushu athletes were also caught in doping scandals, believed to be caused by inadvertent use of banned substances [5].

Inadvertent doping also involves the use of nutritional supplements among athletes. In past studies in Malaysia, 70% of the elite athletes and 40% of the youth athletes were reported to be consuming nutritional supplements [6, 7]. Inadvertent doping results when supplement labels contain misinformation that misleads athletes to falsely believe that prohibited products are allowed in sports [8].

The key to addressing the issue of inadvertent doping by athletes is prevention [9]. Pharmacist, as a healthcare professional that is specifically trained in medication use, can play an important role in combating doping in sports. Athletes may attend to the community pharmacy as a customer or a patient, to purchase medications, over-the-counter medications, or health supplements. For examples, prescribed medication such as salbutamol inhaler is a common medication used for asthma but excessive use without knowing the upper dose limit could violate the doping rules. Some over-the-counter medications may be sold with different active ingredients in different countries. A British skier had his Olympics bronze medal stripped due to unaware of the presence of banned substance in the formulation of the nasal inhaler he bought overseas during competition [10]. Previous literatures also reported that some athletes obtained drug products from pharmacies [11] while others would seek pharmacists' advice on the use of medications or supplements for various conditions, including the management of sports injuries [12]. The International Pharmaceutical Federation (FIP) also recognises pharmacists' roles in preventing doping in sports by updating themselves on the World Anti-Doping Code and help athletes to identify prohibited substances in sports [13].

Although pharmacists are generally well-equipped with knowledge on medications, their abilities and readiness to counsel the athletes are yet to be known. The additional knowledge on the Prohibited List published by World Anti-Doping Agency (WADA) annually is the minimum requirement for the pharmacists to provide correct information to the athletes. Nevertheless, previous studies have reported that most of the pharmacists did not have sufficient knowledge on the prohibited substances in sports [14, 15]. However, the current levels of knowledge among Malaysian community pharmacists about doping in sports are not well studied. This study expands on the survey conducted by Chiang et al in the capital city of Malaysia, Kuala Lumpur [16]. In our study, we have expanded the data collection to other states in Malaysia and assessed other aspects that were not investigated.

Thus, this study aims to offer important insights into the factors that could influence community pharmacists' levels of doping-related knowledge and provides overview of Malaysian community pharmacists' knowledge of doping in sports especially with respect to their readiness to take part in anti-doping initiatives. Assessing their knowledge helps to identify new avenues for future studies and also areas of deficiency that would require interventions to improve pharmacists' current roles in assisting athletes with medication use and avoiding unintentional intake of banned substances.

## Methods

### Study design and sampling

This is a cross-sectional survey conducted online and via distribution of hard copies of questionnaires to community pharmacists in Malaysia. A list of registered community pharmacists was obtained from the Malaysian Pharmaceutical Society's website, and a simple randomised list was generated. The respondents included in this study were pharmacists fully registered with the Pharmacy Board of Malaysia and had been working in the community pharmacy setting for at least one year.

### Data collection

A pilot study was carried out to test the reliability and validity of the questionnaire. The pilot study was done on 31 community pharmacists prior to data collection. Minor improvement was made based on their feedback. Questionnaires were distributed to selected community pharmacists from May 2019 until November 2019. They were briefed about the objectives of the study before consenting to take part. We have included an explanatory statement at the beginning of the questionnaire that advises the respondents not to refer to any resources when answering the knowledge-based questions. This study was approved by the Research Ethics Committee, the National University of Malaysia (UKM PPI/111/8/JEP-2018-215).

### Sample size calculation

To ensure that the study findings generalizable to the whole community pharmacist population in Malaysia, we used the Krejcie and Morgan formula was used to calculate the required sample size [17]. The calculation was based on the total number of community pharmacists in Malaysia, determined to be 3094 [18]. The minimum sample size was calculated at 384.

### Study instruments

The demographic section contained 13 questions about socio-demographic characteristics, namely gender, age, race, place of practice, type of place of practice, practice premises, profession, academic qualification, experience of practising abroad, postgraduate qualification, professional membership, number of years in practice, and understanding of the term 'doping'. We took into accounts of the experience of practicing abroad as improved knowledge on pharmacy practice and patient care is associated with exposure of international experiences [19].

The knowledge section included questions that were adapted from a previous survey that assessed the readiness of community pharmacists as counsellors for athletes [16]. The questions can be classified into five main domains: 1) Prohibited substances in sports; 2) The roles of WADA; 3) Anti-doping rule violations; 4) General knowledge of doping; 5) Doping cases in Malaysia. We wrote several additional questions about the roles of WADA, types of doping violations, athlete biological passport (ABP), and the doping situation in Malaysia, based on The Prohibited List 2019 and World Anti-Doping Code 2019 [20–22]. Overall, this section consists of 10 multiple-choice and true or false questions.

Assessment of the knowledge is done based on the marks that the respondents obtain. A score of 2 was given to each correct answer; 1 for 'not sure'; and 0 for each wrong answer. The total score was 68. The respondents were then grouped into three categories based on their knowledge levels. Respondents who scored 41 or less (60% or less of 68) were classified as having poor levels of knowledge; 42–56 (61%-83% of 68) as having moderate levels of knowledge; and 57 or more (84% or more of 68) as having good levels of knowledge [16, 20].

## Data analysis

The data were analysed using the Statistical Package of Science Analysis (SPSS) version 25. Continuous data was presented as medians alongside values for the interquartile range. The Mann-Whitney U and Kruskal-Wallis tests were used to determine the association between the respondents' demographic characteristics and knowledge scores. A p-value of $<0.05$ was used to mark statistical significance.

## Results

### Pilot study

The reliability and validity of the questionnaire was assessed (Npilot = 31) prior to data collection. The value of Cronbach's alpha obtained for the knowledge section was 0.780, which indicated good reliability.

### Response rate

Over a period of seven months, 426 questionnaires were distributed, and 384 questionnaires were completed and returned, giving a response rate of 90.14%. Overall, the missing value calculated from the valid responses was 0.61%.

### Demographic characteristics

The demographic characteristics of the respondents (N = 384) are shown in Table 1. Most of the respondents were women (N = 244, 63.5%), aged between 23 and 30 (N = 182, 47.4%), and worked in cities or urban areas (N = 297, 78.0%), particularly the Federal Territory of Kuala Lumpur (N = 160, 41.8%).

The majority of the respondents (N = 284, 74.9%) obtained their pharmacy degrees from local universities. Only 13.1% (N = 50) of the respondents had worked abroad, and most of them did so for less than one year (N = 25, 6.5%). A small number (4.2%, N = 16) of the respondents had postgraduate degrees (PhD or master's). Most of the respondents (N = 248, 65.6%) were members of professional bodies, the main one being the Malaysian Pharmaceutical Society (MPS) (N = 241, 62.8%). In terms of their professional experience measured by years in practice, most of the respondents had been practising in various settings for a total of two to five years (N = 147, 39.2%). The median number of years of working as a community pharmacist was six years with the interquartile range (IQR) being nine years.

### The respondents' familiarity of the term 'doping'

More than three-quarters of the respondents (N = 307, 80.6%) had heard of the term 'doping'.

### The respondents' knowledge of doping in sports

Table 2 shows a descriptive analysis of the five domains of the respondents' knowledge of doping in sports.

**Knowledge of prohibited substances in sports.** Most of the respondents knew that anabolic-androgenic steroids (N = 368, 95.8%), stimulants (N = 302, 78.6%), and growth factors (N = 252, 65.6%) are prohibited in sports. These drugs are prohibited at all times in sports. Insulin (N = 31, 8.1%) and beta-blockers (N = 98, 25.5%) were, however, lesser known among the respondents. Alcohol (N = 78, 20.3%) was most frequently mistaken by the respondents as a prohibited substance, followed by caffeine (N = 68, 17.7%), nicotine (N = 58, 15.1%) and NSAIDs (N = 33, 8.6%). We found that 71.6% of the respondents (N = 275) were aware that

**Table 1. The respondents' demographic characteristics (n = 384).**

| Demographic variables | Number (n = 384) | Percentage (%) |
|---|---|---|
| **Gender** | | |
| **Male** | 140 | 36.5% |
| **Female** | 244 | 63.5% |
| **Age** | | |
| **23–30 years** | 182 | 47.4% |
| **31–40 years** | 135 | 35.2% |
| **41–50 years** | 48 | 12.5% |
| **>50 years** | 19 | 4.9% |
| **Race** | | |
| **Malay** | 76 | 19.8% |
| **Chinese** | 279 | 72.7% |
| **Indian** | 22 | 5.7% |
| **Others** | 7 | 1.8% |
| **Place of practice** | | |
| **Northern Peninsular** | | |
| • (Perlis, Kedah, Penang, Perak)     Eastern Peninsular | 79 | 20.6% |
| • (Kelantan, Terengganu, Pahang) Central Peninsular | 24 | 6.3% |
| • (Selangor, Negeri Sembilan) Southern Peninsular | 80 | 20.9% |
| • (Melaka, Johor) East Malaysia | 36 | 9.4% |
| • (Sabah, Sarawak) Federal Territories | 4 | 1.0% |
| • (Labuan, Kuala Lumpur, Putrajaya) | 160 | 41.8% |
| **Type of place of practice** | | |
| **City or urban** | 297 | 78.0% |
| **Town or suburban** | 84 | 22.0% |
| **Academic qualification** | | |
| **Local (Bachelor's degree or equivalent)** | 284 | 74.9% |
| **Foreign (Bachelor's degree or equivalent)** | 87 | 23.0% |
| **Local & Foreign (Bachelor's degree or equivalent)** | 8 | 2.1% |
| **Experience of practising overseas** | | |
| **Yes** | 50 | 13.1% |
| **≤1 year (out of 50)** | 25 | 50.0% |
| **1–5 years (out of 50)** | 18 | 36.0% |
| **>5years (out of 50)** | 3 | 6.0% |
| **Not stated (out of 50)** | 4 | 8.0% |
| **No** | 332 | 86.9% |
| **Presence of postgraduate degree (PhD or Master's degree)** | | |
| **Yes** | 16 | 4.2% |
| **No** | 365 | 95.8% |
| **Involvement as member of professional bodies** | | |
| **Yes** | 248 | 65.6% |
| **Malaysian Pharmaceutical Society (MPS)** | 241 | 62.8% |
| **General Pharmaceutical Council (GPhC)** | 6 | 1.6% |
| **Royal Pharmaceutical Society of Great Britain (RPSGB)** | 6 | 1.6% |
| **Malaysian Community Pharmacy Guild (MCPG)** | 3 | 0.8% |
| **Malaysia Pharmacy Board** | 3 | 0.8% |
| **No** | 130 | 34.4% |
| **Years of practice (median = 6, IQR = 9)** | | |

*(Continued)*

Table 1. (Continued)

| Demographic variables | Number (n = 384) | Percentage (%) |
|---|---|---|
| <2 years | 34 | 9.1% |
| 2–5 years | 147 | 39.2% |
| 6–10 years | 84 | 22.4% |
| 11–20 years | 83 | 22.1% |
| >20 years | 27 | 7.2% |
| **Have you heard about the term 'doping'?** | | |
| Yes | 307 | 80.6% |
| No | 74 | 19.4% |

diuretics can be used by athletes as masking agents; however, only 55.5% (N = 213) knew diuretics are prohibited in sports.

This section also contains questions about the use of drugs in competition. Most of the respondents (N = 272, 70.8%) were able to identify salbutamol as being allowed in competition, but less than half of the respondents were able to do so for salmeterol (N = 172, 44.8%) and inhaled corticosteroids (N = 175, 45.6%). Less than a quarter of the respondents misidentified oral corticosteroids (N = 46, 12.0%), injected corticosteroids (N = 35, 9.1%), and dihydrocodeine (N = 18, 4.7%) as substances allowed in competition.

**Knowledge of the roles of the World Anti-Doping Agency (WADA).** The role of WADA in coordinating anti-doping initiatives worldwide was correctly identified by most of the respondents (N = 314, 81.8%), and this is followed by the other functions the agency serves in publishing updated lists of prohibited substances in sports (N = 293, 76.3%) and establishing the World Anti-Doping Code (N = 251, 65.4%). Most of the respondents (N = 254, 66.1%) knew that WADA does not prosecute athletes who violate anti- doping rules. However, many of them (N = 246, 64.1%) did not know that the tests for detecting prohibited substances in blood or urine samples are not conducted by WADA but WADA- accredited laboratories.

**Knowledge of doping violations.** Most of the respondents were aware that doping violations include the presence of a prohibited substance in a blood or urine sample (N = 352, 91.7%), refusal to undergo a doping test requested by authorised personnel (N = 286, 74.5%), and administration of a prohibited substance to an athlete (N = 276, 71.9%). Slightly over half of the respondents (N = 196, 51.0%) were aware that being complicit in the trafficking of prohibited substances to athletes is also a doping violation. Only a minority of the respondents knew that unintentional intake of prohibited substances by athletes is a doping offense (N = 131, 34.1%).

**General knowledge of doping.** More than half of the respondents (N = 243, 63.3%) knew that the Therapeutic Use Exemption (TUE) allows athletes to use prohibited substances for medical reasons in or out of competition. We found that only 26.0% (N = 100) correctly identified ABP as a programme that monitors selected biological variables over time to indirectly reveal the effects of doping.

**Knowledge of the anti-doping initiatives in Malaysia.** Most of the respondents (N = 364, 94.8%) mistakenly assumed or were unsure whether the professional bodies in Malaysia provided guidelines on the use of prohibited substances in sports. Many of the respondents (N = 376, 97.9%) were still unaware that the Analytical Biochemistry Research Centre, Universiti Sains Malaysia had been removed from WADA's list of accredited laboratories that carry out anti-doping drug testing. Also, a small number of respondents (N = 42, 10.9%) knew that the National Sports Institute in Malaysia is not the official anti-doping agency in Malaysia.

**Table 2. The respondents' knowledge of doping in sports (N = 384).**

| Domains | Variables | Correct answer | Number of respondents with correct answer, N (%) | Number of respondents with the wrong answer / not sure answer, N (%) |
|---|---|---|---|---|
| **Knowledge on prohibited substances in sports** | The substances classified by the World Anti-Doping Agency (WADA) as prohibited in sports include:<br>(i) Anabolic-androgenic steroids (AASs)<br>(ii) Peptide hormones<br>(iii) Growth factors<br>(iv) Beta-2 agonists<br>(v) Insulin<br>(vi) Stimulants<br>(vii) Diuretics<br>(viii) Nicotine<br>(ix) Non-steroidal anti-inflammatory drugs (NSAIDs)<br>(x) Beta-blockers<br>(xi) Caffeine<br>(xii) Alcohol<br>Athletes use diuretics as masking agents to hide the presence of other banned substances in their urine.<br>Which of the following drugs can be used by an athlete in competition only?<br>(i) Salbutamol<br>(ii) Salmeterol<br>(iii) Inhaled corticosteroids<br>(iv) Oral corticosteroids<br>(v) Injected corticosteroids | True<br>True<br>True<br>True<br>True<br>True<br>True<br>False<br>False<br>True<br>False<br>False<br>True<br>True<br>True<br>False<br>False | 368 (95.8)<br>174 (45.3)<br>252 (65.6)<br>132 (34.4)<br>31 (8.1)<br>302 (78.6)<br>213 (55.5)<br>326 (84.9)<br>351 (91.4)<br>98 (25.5)<br>316 (82.3)<br>306 (79.7)<br>275 (71.6)<br>272 (70.8)<br>172 (44.8)<br>175 (45.6)<br>338 (88.0)<br>349 (90.9) | 16 (4.2)<br>210 (54.7)<br>132 (34.4)<br>252 (65.6)<br>353 (91.9)<br>82 (21.4)<br>171 (44.5)<br>58 (15.1)<br>33 (8.6)<br>286 (74.5)<br>68 (17.7)<br>78 (20.3)<br>109 (28.4)<br>112 (29.2)<br>212 (55.2)<br>209 (54.4)<br>46 (12.0)<br>35 (9.1) |
| | (vi) Dihydrocodeine | False | 366 (95.3) | 18 (4.7) |
| **Knowledge on the roles of World Anti-Doping Agency (WADA)** | The roles of the World Anti-Doping Agency (WADA) include:<br>(i) To coordinate anti-doping activities worldwide.<br>(ii) To establish the World Anti-Doping Code.<br>(iii) To establish a list of prohibited substances in sports.<br>(iv) To conduct tests for prohibited substances in blood or urine samples.<br>(v) To prosecute doping offenders in sports. | True<br>True<br>True<br>False<br>False | 314 (81.8)<br>251 (65.4)<br>293 (76.3)<br>138 (35.9)<br>254 (66.1) | 70 (18.2)<br>133 (34.6)<br>91 (23.7)<br>246 (64.1)<br>130 (33.9) |
| **Knowledge on doping violations** | Doping violations include:<br>(i) Presence of a prohibited substance in blood or urine.<br>(ii) Helping in trafficking prohibited substances to athletes.<br>(iii) Refusing to undergo a doping test requested by authorized personnel.<br>(iv) Administering a prohibited substance to an athlete.<br>(v) Unintentional intake of a prohibited substance. | True<br>True<br>True<br>True<br>True | 352 (91.7)<br>196 (51.0)<br>276 (71.9)<br>286 (74.5)<br>131 (34.1) | 32 (8.3)<br>188 (49.0)<br>108 (28.1)<br>98 (25.5)<br>253 (65.9) |
| **General knowledge of doping scenarios** | Therapeutic Use Exemption (TUE) allows athletes to use prohibited substances for medical reasons in or out of competition. | True | 243 (63.3) | 141 (36.7) |
| | The Athlete Biological Passport (ABP) is a program that monitors selected biological variables over time to indirectly reveal the effects of doping rather than attempting to detect the doping substance or method itself. | True | 100 (26.0) | 284 (74.0) |
| **Knowledge on doping in Malaysia** | Do you know whether your professional body (e.g Malaysian Medical Council, Malaysian Pharmaceutical Society, etc.) has a guideline on the use of prohibited substances in sports? | False | 20 (5.2) | 364 (94.8) |
| | Universiti Sains Malaysia, Penang (USM) Analytical Biochemistry Research Centre (ABrC), formerly known as Doping Control Centre (DCC), has WADA accreditation to carry out anti-doping drug testing. | False | 8 (2.1) | 376 (97.9) |
| | The National Sports Institute in Malaysia is the official anti-doping agency in Malaysia. | False | 42 (10.9) | 342 (89.1) |

**Table 3. The respondents' levels of knowledge of doping (N = 384).**

| Total score | Number (N = 384) | Percentage (%) |
|---|---|---|
| (median = 52, IQR = 6) | | |
| ≤41 (Poor) | 10 | 2.6% |
| 42–56 (Moderate) | 333 | 86.7% |
| ≥57 (Good) | 41 | 10.7% |

**Total scores of the respondents' doping-related knowledge.** Based on their total knowledge scores, the respondents were classified as having poor, moderate, or good levels of knowledge of doping in sports (Table 3).

Only 1.8% (N = 7) of the respondents scored 41 or less (poor levels of knowledge). More than three-quarters of the respondents (N = 317, 82.6%) scored between 42 to 56 (moderate levels of knowledge). A quarter of the respondents (N = 60, 15.6%) scored 57 or more (good levels of knowledge). The median score was 52 with an IQR of 6.

## Statistical studies

Table 4 shows the relationship between the respondents' demographic characteristics and knowledge scores.

The respondents were grouped based on their demographic characteristics. The Mann-Whitney U test (U value) was used when the comparison of mean ranks (knowledge scores) involved only two groups of respondents, while the Kruskal-Wallis test (H value) was used to compare mean ranks between more than two groups of respondents. Through these statistical analyses, we found the knowledge scores to be significantly affected by the respondents' postgraduate qualifications (or lack thereof) (p = 0.047), professional membership (p = 0.001), amount of professional experience measured in years (p = 0.045), and understanding of the term 'doping' (p<0.0001).

As shown in Table 5, those who with postgraduate degrees (N = 16) had a significantly larger mean rank than those without any postgraduate qualifications (N = 365; 244.50 vs 188.65, U = 2064, p = 0.047). The respondents who were members of professional bodies (N = 248) had a significantly larger mean rank than those without any professional membership (N = 130; 203.26 vs 163.25, U = 12707, p = 0.001). The respondents who had been practicing for ≥6 years i.e., 6–10 years (N = 84), 11–20 years (N = 83), and >20 years (N = 83) had significantly higher mean ranks than those with less experience i.e., <2 years (N = 34) and 2–5 years in practice (N = 147; 210.54, 194.09, and 211.30 vs 179.28 and 169.42, H = 9.735, p = 0.045). The respondents who understood the term 'doping' well (N = 307) had a larger mean rank than those who did not (N = 74; 210.08 vs 111.85, U = 5502, p<0.0001).

**Table 4. Association between demographic variables and the respondents' knowledge scores.**

| | Gender | Race | Place of practice | Academic qualifications | Experienc e of practicing overseas | Presence of postgraduat e degree | Involvemen t as a member of professional bodies | Years of practice | Familiarit y with term 'doping' |
|---|---|---|---|---|---|---|---|---|---|
| | p-value (p<0.05) | | | | | | | | |
| Total score on knowledge | 0.559[a] | 0.821[b] | 0.987[b] | 0.971[b] | 0.435[a] | **0.047[a]** | **0.001[a]** | **0.045[b]** | **0.000[a]** |

[a] Mann-Whitney U test.

[b] Kruskal-Wallis test.

**Table 5. Comparison of mean rank between associated demographic variables and total score of community pharmacists' knowledge.**

| Associated demographic variables | Statistical tests result (p<0.05) | Components of associated demographic variables | Mean rank |
|---|---|---|---|
| **Presence of postgraduate degree** | U = 2064, p = 0.047 | With postgraduate degree (N = 16) | 244.50 |
| | | Without a postgraduate degree (N = 365) | 188.65 |
| **Involvement as member of professional bodies** | U = 12707, p = 0.001 | Member (N = 248) | 203.26 |
| | | Non-member (N = 130) | 163.25 |
| **Years of practice** | H = 9.735, p = 0.045 | <2 years (N = 34) | 179.28 |
| | | 2–5 years (N = 147) | 169.42 |
| | | 6–10 years (N = 84) | 210.54 |
| | | 11–20 years (N = 83) | 194.09 |
| | | >20 years (N = 27) | 211.30 |
| **Familiarity with term 'doping'** | U = 5502, p = 0.000 | Familiar (N = 307) | 210.08 |
| | | Unfamiliar (N = 74) | 111.85 |

U = Mann-Whitney U test; H = Kruskal-Wallis test

## Discussion

The current study evaluated the Malaysian community pharmacists' knowledge related to drugs in sports. In general, most of the respondents had heard of the term 'doping' and were able to describe it adequately as the use or misuse of drugs by athletes to enhance their performance in sports. Most of them were able to identify anabolic-androgenic steroids (AAS), stimulants, and growth factors as prohibited substances in sports. This is consistent with previous studies, which reported that pharmacists were able to identify anabolic-androgenic steroids and stimulants as prohibited substances in sports [23, 24]. We also found that most of the respondents knew that diuretics could be used as masking agents, a finding similarly reported by Chiang et al [16]. The pharmacists' familiarity with these substances could be associated with the popularity of the substances in doping cases and they were the most abused substances in sports as proven by the reports by WADA in 2017 stating that up to 58% of all adverse analytical findings in doping tests came from AAS and stimulants [25].

However, most of the respondents failed to identify insulin and beta-blockers as prohibited substances. This is because beta blockers are prohibited in-competition for certain sports only [21]. So, the respondents might have lesser awareness on these unpopular doping substances. Additionally, WADA reported in 2017 that only 0.3% of doping tests were positive for beta-blockers, suggesting that the drugs were infrequently misused by athletes [25]. This may explain why these substances were less recognised by the respondents as prohibited substances. Meanwhile, the respondents might not be aware of the mechanism and the reason of insulin being used as a doping agent. Insulin is normally used by diabetic patients for treating high sugar level, but it could be misused by bodybuilders and weightlifters to suppress proteolysis and increase protein synthesis for faster muscle gain [26].

Besides, the study extends our knowledge on the familiarity of the Malaysian community pharmacists on the definition of doping violations. Most of the respondents in this study were unaware that helping in trafficking prohibited substances to athletes and unintentional intake of a prohibited substance are also considered doping violations. The results show the lack of awareness of the community pharmacists on the doping definition published by WADA which clearly states that athletes should be responsible for everything they ingest, and even accidental intake of banned substance would violate the doping rules [22]. Therefore, pharmacists need to step up in expanding their knowledge so that in the future they could advise the athletes and become their support personnel in building a healthy and sustainable sports career for them.

Only 63.3% of the respondents in this study knew that TUEs are required for the use of drugs by athletes, and the proportion is lower than the 75.9% reported by Chiang et al [16]. In comparison, 45.2% of the South African pharmacists who responded to a survey scored poorly on the knowledge regarding TUEs [27]. These findings indicate that there is still a need for educating pharmacists on the importance of TUEs as a mechanism that enables the use of prohibited substances or methods in the treatments of illnesses, injuries, or chronic medical conditions experienced by athletes [28]. Most of the respondents in this study were also unaware that ABP is a newly introduced doping detection method. ABP is relatively simple and can be potentially adopted by many countries as an effective measure against doping [29]. A full understanding of the harmonised modules employed in ABP, including the haematological and steroidal modules, is the first step towards establishing proper ABP testing facilities. The ABP Operating Guidelines [30], published by WADA, harmonize both modules and are a good resource for establishing proper ABP testing facilities.

Furthermore, almost 90% of the respondents in our study did not know the doping initiatives and official bodies in Malaysia. Malaysia once had an accredited laboratory at Universiti Sains Malaysia, which was suspended due to non-compliance with the International Standard for Laboratories [31]. Besides, most of the community pharmacists did not know that the official anti-doping agency in Malaysia that is tasked with fighting doping is Anti-Doping Agency of Malaysia (ADAMAS) despite its existence since 2007 [32]. The percentage is substantially lower than that reported in another study i.e., 54.9% of Slovenian pharmacists knew their national anti-doping agency [20]. Failure to recognize the proper source of information to refer to when meeting athletes in their working environment may make pharmacists unable to provide correct recommendations and advice to the athletes. The Irish College General Practitioners (ICGP) published guidelines to educate general practitioners on doping-related regulations and their roles and involvement in the prevention of doping in sports. The guidelines are reviewed periodically, with the latest edition being published in 2015 [33]. In contrast, no professional bodies in Malaysia have published guidelines on the use of prohibited substances in sports; but most of the respondents in our study were unaware of this. Professional bodies in every country, including Malaysia, should adopt a similar practice to ICGP and publish guidelines for engaging healthcare professionals in the prevention of doping in sports.

Overall, our study demonstrated that the average knowledge score of Malaysian community pharmacist on doping was moderate. This is in line with previous literatures by Lemettilä et al (2021) and Gebregers et al (2021) [15, 34]. These findings pointed out the needs to improve pharmacists' knowledge in drugs in sports which could be done by establishing courses on drugs in sports during university study or special courses on drugs in sports for the working pharmacists. In Malaysia, subjects related to doping in sports are incorporated in curriculum in pharmacy programmes either as a core subject with three credit hours or elective subject with two credit hours. However, some universities in Malaysia did not offer the subjects to their students [35]. The lack of exposure and training provided during university may then cause the pharmacists to have low confidence when they are dealing with issues related to prohibited substances in sports.

The correlation analysis showed that the respondents with six or more years of professional experience had better knowledge and could potentially be trained to become drug advisors or counsellors for athletes. A good understanding of the term 'doping' predicted better knowledge scores. Thus, pharmacists should be encouraged to learn more about doping-related issues. We found that the respondents who were members of professional bodies obtained significantly better knowledge scores. Most of the respondents in this study were members of Malaysian Pharmaceutical Society. This national association for pharmacists periodically organises courses and seminars for professional development, averaging ~15 programmes per

month in the past three years [36]. Thus, pharmacists that are interested in getting more information on drugs in sports should take self-initiative to attend relevant courses offered by anti-doping agency to keep themselves up to date.

Sports pharmacy is considered a relatively new and emerging fields especially in South-East Asia. Pharmacists are traditionally perceived by the public as specialists in medication dispensing and counselling. However, their role of pharmacist in healthcare has expanded over the years towards primary prevention through health education. Athletes are a special population with generally good health status but may still consume a relatively large number of medications and supplements compared to the ordinary healthy individuals. Thus, it is important for pharmacists to engage and provide their professional service to athletes in the future to eliminate inadvertent doping.

## Limitations

The first limitation of the study relates to the use of self-administered questionnaires. Although the explanatory statement clearly indicates that the respondents should answer the knowledge-based questions honestly without referring to any resources, some might not have followed the guidelines. This may have led to inaccuracy in the assessment of the respondents' knowledge. Second, the respondents were recruited through convenience sampling, and the questionnaires were not evenly distributed to community pharmacists in the different states of Malaysia. Thus, the results of this study may not be generalised to the entire population of community pharmacists across Malaysia.

## Conclusion

We found that community pharmacists in Malaysia had moderate levels of doping- related knowledge. They were able to identify prohibited substances commonly misused by athletes. Most were still unaware that inadvertent doping constitutes a doping violation, despite its being the primary contributor to the prevalence of doping in sports. Most were also ill-informed about the doping situation in Malaysia. Hence, more doping-related programmes and activities should be organised to enhance community pharmacists' knowledge of inadvertent doping and transform them into proactive participants in contemporary anti-doping initiatives.

## Supporting information

**S1 Table. Supporting information for respondents' demographic characteristics.**
(XLSX)

**S2 Table. Supporting information for respondents' levels of knowledge in doping.**
(XLSX)

**S3 Table. Supporting information for demographic variables and the respondents' knowledge scores.**
(XLSX)

## Acknowledgments

We would like to express our gratitude to Anti-Doping Agency of Malaysia (ADAMAS) and Malaysian Pharmaceutical Society (MPS) and the willingness of the community pharmacists to be involved as respondents.

## Author Contributions

**Conceptualization:** Eng Wee Chua, Tuan Mazlelaa Tuan Mahmood, Sharifa Ezat Wan Puteh, Jyh Eiin Wong, Ahmad Taufik Jamil, Jamia Azdina Jamal, Ahmad Fuad Shamsuddin, Adliah Mhd Ali.

**Data curation:** Nontharit Voravuth.

**Formal analysis:** Nontharit Voravuth.

**Funding acquisition:** Ahmad Fuad Shamsuddin.

**Investigation:** Nontharit Voravuth, Eng Wee Chua, Tuan Mazlelaa Tuan Mahmood, Ming Chiang Lim, Sharifa Ezat Wan Puteh, Nik Shanita Safii, Jyh Eiin Wong, Ahmad Taufik Jamil, Jamia Azdina Jamal, Ahmad Fuad Shamsuddin, Adliah Mhd Ali.

**Methodology:** Eng Wee Chua, Tuan Mazlelaa Tuan Mahmood, Sharifa Ezat Wan Puteh, Nik Shanita Safii, Jyh Eiin Wong, Ahmad Taufik Jamil, Jamia Azdina Jamal, Ahmad Fuad Shamsuddin, Adliah Mhd Ali.

**Project administration:** Eng Wee Chua, Ahmad Fuad Shamsuddin, Adliah Mhd Ali.

**Resources:** Ming Chiang Lim, Ahmad Fuad Shamsuddin.

**Supervision:** Eng Wee Chua, Sharifa Ezat Wan Puteh, Jyh Eiin Wong, Jamia Azdina Jamal, Ahmad Fuad Shamsuddin, Adliah Mhd Ali.

**Validation:** Eng Wee Chua, Sharifa Ezat Wan Puteh, Jyh Eiin Wong, Ahmad Taufik Jamil, Jamia Azdina Jamal, Ahmad Fuad Shamsuddin, Adliah Mhd Ali.

**Visualization:** Eng Wee Chua, Sharifa Ezat Wan Puteh, Jyh Eiin Wong, Ahmad Taufik Jamil, Jamia Azdina Jamal, Ahmad Fuad Shamsuddin, Adliah Mhd Ali.

**Writing – original draft:** Nontharit Voravuth.

**Writing – review & editing:** Eng Wee Chua, Ming Chiang Lim, Jyh Eiin Wong, Adliah Mhd Ali.

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
