## [Decision Letter · Decision Letter 0]

1 Sep 2021

PONE-D-21-01079

Engaging Community Pharmacists to eliminate Inadvertent Doping in Sports: A Study of their Knowledge on Doping

PLOS ONE

Dear Dr. Ali,

Thank you for submitting your manuscript to PLOS ONE. After careful consideration, we feel that it has merit but does not fully meet PLOS ONE’s publication criteria as it currently stands. Therefore, we invite you to submit a revised version of the manuscript that addresses the points raised during the review process.

We look forward to receiving your revised manuscript.

Kind regards,

Muhammad Shahzad Aslam, Ph.D.,M.Phil., Pharm-D

Academic Editor

PLOS ONE

Journal Requirements:

“We would like to express our gratitude to UNESCO for funding this research. This survey would not have been possible without the assistance from Anti-Doping Agency of Malaysia (ADAMAS) and Malaysian Pharmaceutical Society (MPS) and the willingness of the community pharmacists to be involved as respondents.”

“This study was supported by a research grant of UNESCO Anti-Doping Fund (4500340762-A2) and Universiti Kebangsaan Malaysia (UKM-NF-2018-001).”

5. Please include a caption for figure 1.

6. Please include your tables as part of your main manuscript and remove the individual files. Please note that supplementary tables (should remain/ be uploaded) as separate "supporting information" files.

Reviewers' comments:

Reviewer's Responses to Questions

**Comments to the Author**

1. Is the manuscript technically sound, and do the data support the conclusions?

Reviewer #1: Yes

Reviewer #2: Yes

2. Has the statistical analysis been performed appropriately and rigorously? 

Reviewer #1: Yes

Reviewer #2: Yes

3. Have the authors made all data underlying the findings in their manuscript fully available?

Reviewer #1: Yes

Reviewer #2: Yes

4. Is the manuscript presented in an intelligible fashion and written in standard English?

Reviewer #1: No

Reviewer #2: Yes

5. Review Comments to the Author

Reviewer #1: The article is interesting and relevant. However, it lacks of rational and the results are very descriptive while they could be linked to hypothesizes. Another weakness is the unbalanced sampling regarding for example age, gender or academic levels. Indeed it’s unclear if this sample is representative of the Malaysian pharmacists.

The introduction is too short and the research question is the question is insufficiently problematized: It is not clear why pharmacists are in the first line regarding doping ? The examples provided in the beginning of the MS did not indicate that the pharmacists were involved in inadvertent doping. Another key point is the cultural definition of the pharmacists wok in Malaysian that is not defined. Indeed, it is expected the potential role of the pharmacists could change between countries: from physician collaboration, patient counselling to commercial benefits… Please explain how pharmacists work in Malaysian: are there able to sell drugs without physicians order? Is there durgs that are free to sell even they are prohibited in the WADA list? Is there a commercial activity linked to the price of drugs that offers potential differential benefits between pharmacists’ office? There is no review of research and most of the references are only proposed in the discussion. In addition, the question of assessment of the knowledge is not considered: what types of knowledge? How they could be assessed ? In the result section, the sample has been split and comparisons using different variables such academic level, year of practice and so on. The reason why these comparisons have been done and are relevant should be explain in the introduction. Are there Hypothesizes for that regarding previous research? In addition in the introduction, it will be also interesting to indicate their potential role for doping violation rule regarding trafficking of prohibited substances to athletes and unintentional intake of a prohibited substances.

To resume, this article did not provide a sufficient report of what you know now and what we need to know that justify this study.

The results are however interesting and the discussion and conclusion clear. Please see if you can also adjust the sample or justify its representativeness.

Reviewer #2: Introduction: Written well.

Methods:

Data Collection: How authors ensured that participants have not referred internet while giving knowledge based questions?

Discussion: Subheading may not requires under discussion.

Conclusion: Please remove the limitations which are mentioned under the heading of conclusion. Make separate heading for limitation.

References: Internet based references please make sure are active and referenced in appropriate manner, if reader would like to visit internet based links. these internet based references should be accessible.

6. PLOS authors have the option to publish the peer review history of their article (what does this mean?). If published, this will include your full peer review and any attached files.

Reviewer #1: No

Reviewer #2: No

---

## [Author Response · Author response to Decision Letter 0]

1 Feb 2022

Dear Editor,

Manuscript: Engaging Community Pharmacists to Eliminate Inadvertent Doping in Sports: A Study on their Knowledge on Doping

We thank the reviewers for their constructive comments. We have revised and proofread the manuscript accordingly. Important changes are annotated with ‘comments’ in the manuscript. Additions are highlighted in yellow; other types of edits are marked with ‘Track Changes’. 

Reviewer #1: 

1. The article is interesting and relevant. However, it lacks of rational and the results are very descriptive while they could be linked to hypothesizes. 

Response to reviewer: The main aim of the study is to gauge the current level of knowledge of doping among community pharmacists. This will in turn inform the design of future studies involving specific interventions among pharmacists based on proven hypotheses. Thus, the descriptive nature of the study is inevitable, as there is a lack of baseline data to form a hypothesis. 

2. In the result section, the sample has been split and comparisons using different variables such academic level, year of practice and so on. The reason why these comparisons have been done and are relevant should be explain in the introduction. Are there Hypothesizes for that regarding previous research? 

Response to reviewer: We have expanded the introduction and added studies describing the effects of prior training on doping-related knowledge levels: “Previous studies reported that specific training, such as university courses and workshops, significantly influenced respondents’ levels of doping-related knowledge [1,2,3].” 

The findings of these studies suggest that academic qualifications or years of experience may affect one’s level of doping-related knowledge. We have also included the findings of these studies in the discussion. 

3. Another weakness is the unbalanced sampling regarding for example age, gender or academic levels. Indeed it’s unclear if this sample is representative of the Malaysian pharmacists.

Response to reviewer: We have expanded the description of the methods to include the study design and the sampling procedure. Also, we have added descriptions of the findings of other studies showing that the demographic characteristics reported by those studies are similar to ours. 

“In terms of the demographic characteristics, we found that the majority of the respondents were young Chinese women with professional degrees in pharmacy. This is in line with the characteristics of the participants enrolled in other studies involving community pharmacists in Malaysia [4,5,6].”

4. The introduction is too short and the research question is insufficiently problematized: It is not clear why pharmacists are in the first line regarding doping? The examples provided in the beginning of the MS did not indicate that the pharmacists were involved in inadvertent doping. 

Response to reviewer: We have added descriptions of the roles of community pharmacists in tackling issues related to doping in sport:

“According to Tsarouhas et al., 15% of athletes obtained drug products from pharmacies [7]. Athletes may also ask pharmacists for advice on the use of medications or supplements for various conditions, including the management of sports injuries [8]. The International Pharmaceutical Federation (FIP) recognises pharmacists’ roles in preventing doping in sports. It is recommended that pharmacists should update themselves on the World Anti-Doping Code and help athletes to identify products containing substances prohibited by the World Anti-Doping Agency (WADA) [9].”

5. Another key point is the cultural definition of the pharmacists work in Malaysian that is not defined. Indeed, it is expected the potential role of the pharmacists could change between countries: from physician collaboration, patient counselling to commercial benefits… Please explain how pharmacists work in Malaysian: are there able to sell drugs without physicians order? Is there drugs that are free to sell even they are prohibited in the WADA list? 

Response to reviewer: In Malaysia, there are different categories of medications that can be dispensed by community pharmacists i.e.:

● Group A Can only be sold by a licensed wholesaler to a pharmacist or to another licensed wholesaler or by a licensed wholesaler to be immediately exported to a purchaser outside Malaysia. 

● Group B Can be dispensed only against prescription of a Registered Medical Practitioner, Dentist, or Veterinary Surgeon, as the case may be and with the prescription in the correct form as required by the law. 

● Group C Can only be sold as a dispensed medicine with entry in the Prescription Book.

● Group D Can only be sold as a dispensed medicine with an entry in the Poisons Book. 

● Part II Poisons Retail sale restricted to Poison Licence Holder. Labelling requirements only.

● NP- Non-scheduled Poisons Non-scheduled poisons or over the counter products for retail sale.

Source: MIMS Malaysia (https://www.mims.com/malaysia/viewer/html/poisoncls.htm)

It depends on:

● Whether the drugs have legitimate medical uses. 

● The potential of the drugs to be misused by athletes to gain unfair advantages in sports. 

Note: This information is not included in the manuscript because it is not directly related to the study findings.

In the original version of the manuscript, we have already highlighted in the discussion the misuse of different classes of drugs that could be prevented by interventions by community pharmacists: 

However, it is worth pointing out that many of the WADA-prohibited drugs are commonly used in medical treatments and are therefore not banned in these settings e.g., insulin is not well known for its performance-enhancing properties, while beta-blockers are prohibited in-competition for only certain sports such as archery and shooting, which may explain why these substances were less recognised by pharmacists as prohibited substances.

6. Is there a commercial activity linked to the price of drugs that offers potential differential benefits between pharmacists’ office? There is no review of research and most of the references are only proposed in the discussion. 

Response to reviewer: This is an interesting point. However, the commercial activity linked to the price is irrelevant to this study. 

7. In addition, the question of assessment of the knowledge is not considered: what types of knowledge? How they could be assessed? 

Response to reviewer: We have expanded the description of how we adapted the questionnaire from previous studies:

“The knowledge section included questions that were adapted from a previous survey that assessed the readiness of community pharmacists as counsellors for athletes [10]. The questions can be classified into five main domains [1,11,12]: 1) Prohibited substances in sports; 2) The roles of WADA; 3) Doping violations; 4) General knowledge of doping; 5) Doping cases in Malaysia.”

8. In addition in the introduction, it will be also interesting to indicate their potential role for doping violation rule regarding trafficking of prohibited substances to athletes and unintentional intake of a prohibited substances.

Response to reviewer: This is an interesting point. However, community pharmacists play no roles in handling anti-doping rule violations. So, this is irrelevant to this study.

In regard to unintentional intake of prohibited substances, we have added the following into our introduction:

“Athletes may become ill or injured, or suffer from chronic medical conditions [13], which may necessitate the use of medications that are banned in sports. For instance, in Japan, many over-the-counter medications contain prohibited substances; this explains the high rate of unintentional doping in the country [14].”

“Inadvertent doping also involves the use of nutritional supplements among athletes. In prior studies, 55% to 98% of athletes were reported to be consuming nutritional supplements [15,16]. Inadvertent doping results when supplement labels contain misinformation that misleads athletes to falsely believe that prohibited products are allowed in sports [17].

9. To resume, this article did not provide a sufficient report of what you know now and what we need to know that justify this study.

Response to reviewer: Referring to our first response, the aim of the study is to obtain baseline findings that will inform the design of future studies involving specific interventions among pharmacists.

Reviewer #2: 

10. Introduction: Written well.

Response to reviewer: Noted with thanks.

11. Methods:

Data Collection: How authors ensured that participants have not referred internet while giving knowledge-based questions?

Response to reviewer: In the respondent explanatory statement, we remind the respondents that their honesty in answering the questionnaire is crucial to ensuring the validity of the study findings. We conducted the study based on ‘trust’ and have faith in pharmacists’ professionalism and ethical integrity. 

Nevertheless, this is an important point. We have briefly discussed it under ‘Limitations’. 

“The first limitation of the study relates to the use of self-administered questionnaires. Although the explanatory statement clearly indicates that the respondents should answer the knowledge-based questions honestly without referring to any resources, some might not have followed the guidelines. This may have led to inaccuracy in the assessment of the respondents’ knowledge.”

12. Reviewer #2: Discussion: Subheading may not requires under discussion. 

Response to reviewer: All the subheadings were removed. 

13. Reviewer #2: Conclusion: Please remove the limitations which are mentioned under the heading of conclusion. Make separate heading for limitation.

Response to reviewer: We have created a separate subsection for the study limitations. 

14. Reviewer #2: References: Internet based references please make sure are active and referenced in appropriate manner, if reader would like to visit internet based links. these internet based references should be accessible. 

Response to reviewer: The references have been amended accordingly.

References cited in this letter:

1. Auersperger I, Topic MD, Maver P, Pusnik VK, Osredkar J, Lainscak M. Doping awareness, views, and experience: a comparison between general practitioners and pharmacists. Wien Klin Wochenschr. 2012;124(1-2):32-8.

2. Howard MS, DiDonato KL, Janovick DL, Schroeder MN, Powers MF, Azzi AG, Lengel AJ. Perspectives of athletes and pharmacists on pharmacist-provided sports supplement counseling: An exploratory study. Journal of the American Pharmacists Association. 2018 Jul 1;58(4):S30-6.

3. Starzak, D., Derman, W., Mckune, A. & Semple, S. 2016. Anti-Doping Knowledge and Opinions of South African Pharmacists and General Practitioners. Journal of Sports Medicine and Doping Studies 6: 1-7.

4. Teong WW, Ng YK, Paraidathathu T, Chong WW. Job satisfaction and stress levels among community pharmacists in Malaysia. Journal of Pharmacy Practice and Research. 2019 Feb;49(1):9-17.

5. Selvaraj A, Redzuan AM, Hatah E. Community pharmacists’ perceptions, attitudes and barriers towards pharmacist-led minor ailment services in Malaysia. International journal of clinical pharmacy. 2020 Apr;42(2):777-85.

6. Francis J, Toh LS, Sellappans R, Loo JS. Awareness of osteoporosis risk assessment tools and screening recommendations among community pharmacists in Malaysia. International journal of clinical pharmacy. 2021 Jan 28:1-9.

7.Tsarouhas K, Kioukia–Fougia N, Papalexis P, Tsatsakis A, Kouretas D, Bacopoulou F, Tsitsimpikou C. Use of nutritional supplements contaminated with banned doping substances by recreational adolescent athletes in Athens, Greece. Food and chemical toxicology. 2018 May 1;115:447-50.

8. Howard, M. S., Didonato, K. L., Janovick, D. L., Schroeder, M. N., Powers, M. F., Azzi, A. G. & Lengel, A. J. 2018. Perspectives of Athletes and Pharmacists on Pharmacist-Provided Sports Supplement Counseling: An Exploratory Study. J Am Pharm Assoc (2003) 58(4s): S30-S36.e32.

9. Awaisu, A., Mottram, D., Rahhal, A., Alemrayat, B., Ahmed, A., Stuart, M. & Khalifa, S. 2015. Research Knowledge and Perceptions of Pharmacy Students in Qatar on Anti-Doping in Sports and on Sports Pharmacy in Undergraduate Curricula. 79.

10. Chiang LM, Hatah E, Shamsuddin AF. The readiness of community pharmacists as counsellors for athletes in addressing issues of the use and misuse of drugs in sports. Lat. Am. J. Pharm. 2018 Jan 1;37(5):1049-55.

11. The World Anti Doping Code: The 2019 Prohibited List, (2019).

12. WORLD ANTI-DOPING CODE 2015 with 2019 amendments, (2019).

13. Overbye, M. & Wagner, U. 2013. Between Medical Treatment and Performance Enhancement: An Investigation of How Elite Athletes Experience Therapeutic Use Exemptions. 24.

14. Shibata, K., Ichikawa, K. & Kurata, N. 2017. Knowledge of Pharmacy Students About Doping, and the Need for Doping Education: A Questionnaire Survey. BMC Res Notes 10(1): 396.

15. Balaravi, B., Mei Qi, C., Wen Jin, C., Quah Lw, S., Ramadas, A. & Karppaya, H. 2017. Knowledge and Attitude Related to Nutritional Supplements and Risk of Doping among National Elite Athletes in Malaysia. 23.

16. Somerville, S. J., Lewis, M. & Kuipers, H. 2005. Accidental Breaches of the Doping Regulations in Sport: Is There a Need to Improve the Education of Sportspeople? Br J Sports Med 39(8): 512-516; discussion 516.

17. Davar, V. 2012. Nutritional Knowledge and Attitudes Towards Healthy Eating of College-Going Women Hockey Players. 37.

Thank you.

Sincerely,

Adliah Mhd Ali

---

## [Decision Letter · Decision Letter 1]

7 Mar 2022

PONE-D-21-01079R1Engaging Community Pharmacists to Eliminate Inadvertent Doping in Sports: A Study of their Knowledge on DopingPLOS ONE

Dear,

Thank you for submitting your manuscript to PLOS ONE. After careful consideration, we feel that it has merit but does not fully meet PLOS ONE’s publication criteria as it currently stands. Therefore, we invite you to submit a revised version of the manuscript that addresses the points raised during the review process. Please submit your revised manuscript by Apr 21 2022 11:59PM. If you will need more time than this to complete your revisions, please reply to this message or contact the journal office at plosone@plos.org. Please include the following items when submitting your revised manuscript:A rebuttal letter that responds to each point raised by the academic editor and reviewer(s). You should upload this letter as a separate file labeled 'Response to Reviewers'.A marked-up copy of your manuscript that highlights changes made to the original version. You should upload this as a separate file labeled 'Revised Manuscript with Track Changes'.An unmarked version of your revised paper without tracked changes. You should upload this as a separate file labeled 'Manuscript'.If applicable, we recommend that you deposit your laboratory protocols in protocols.io to enhance the reproducibility of your results. Protocols.io assigns your protocol its own identifier (DOI) so that it can be cited independently in the future. For instructions see: https://journals.plos.org/plosone/s/submission-guidelines#loc-laboratory-protocols. Additionally, PLOS ONE offers an option for publishing peer-reviewed Lab Protocol articles, which describe protocols hosted on protocols.io. Read more information on sharing protocols at https://plos.org/protocols?utm_medium=editorial-email&utm_source=authorletters&utm_campaign=protocols.

We look forward to receiving your revised manuscript.

Kind regards,

Muhammad Shahzad Aslam, Ph.D.,M.Phil., Pharm-D

Academic Editor

PLOS ONE

Reviewers' comments:

Reviewer's Responses to Questions

**Comments to the Author**

1. If the authors have adequately addressed your comments raised in a previous round of review and you feel that this manuscript is now acceptable for publication, you may indicate that here to bypass the “Comments to the Author” section, enter your conflict of interest statement in the “Confidential to Editor” section, and submit your "Accept" recommendation.

Reviewer #3: (No Response)

Reviewer #4: (No Response)

Reviewer #5: All comments have been addressed

2. Is the manuscript technically sound, and do the data support the conclusions?

Reviewer #3: Yes

Reviewer #4: Yes

Reviewer #5: Partly

3. Has the statistical analysis been performed appropriately and rigorously? 

Reviewer #3: Yes

Reviewer #4: I Don't Know

Reviewer #5: Yes

4. Have the authors made all data underlying the findings in their manuscript fully available?

Reviewer #3: Yes

Reviewer #4: Yes

Reviewer #5: Yes

5. Is the manuscript presented in an intelligible fashion and written in standard English?

Reviewer #3: Yes

Reviewer #4: Yes

Reviewer #5: Yes

6. Review Comments to the Author

Reviewer #3: The manuscript describes the doping problem at hand in Malaysia and the reason why community pharmacist engagement is crucial in combating the problem. The reviewer queries have been adequately addressed. The only fault I found was with a sentence in the abstract which needs correction. Please fix this at line 9 of the abstract. "With median years of practising as a community pharmacist for six years". The "for" in this phrase must be replaced with "of".

Reviewer #4: Thank you for inviting me to review this manuscript. Overall, I do believe that the manuscript provides interesting and relevant information that is crucial to make further recommendations on how pharmacists can be involved in mitigating inadvertent doping amongst athletes.

I did read through comments from Reviewer 1 and Reviewer 2 and responses from the authors. While I feel that the comments were adequately addressed, I had a few other comments:

1. The introduction was significantly improved. However, I believe the background, significance, and rationale of the study can be even more improved. I would suggest that the authors discuss specifically medications that are often associated with inadvertent doping to give some context. This should be categorized into over-the-county versus prescription drugs.

2. I also believe that a clear description of the roles of community pharmacists should be clearly mentioned, potentially in a section under Setting to provide some contextual understanding about how medications are dispensed (with or without a prescription), etc. In essence, response to Review #1, question 5, should be inserted into the text.

3. It is still quite unclear to me how community pharmacists will be directly involved to eliminate inadvertent doping based on the data presented. While you have pointed out that more education can be provided, what practical tasks are you recommending that pharmacists do? For example, if someone comes into the pharmacy, are there standardized questionnaire to identify that (1) this is an athlete, and (2) they may be using drugs that lead to intentional or inadvertent doping. What interventions exist in the community pharmacy literature that have been shown to have impact? This is still very unclear to me... If you are to keep the study title as is: "Engaging Community Pharmacists to Eliminate Inadvertent Doping in Sports: A Study of their Knowledge on Doping", the "engaging" component will need additional justification and literature support.

Reviewer #5: Congratulation on your work. I do have some comments and questions about your research.

Introduction

1. The rationale of this study. You already mentioned that Chiang et al. had surveyed on this particular subject. What is the difference between the previous survey and yours? If you emphasize this point, it will make the gap or rationale of your study much more transparent.

2. You state that your study highlights community pharmacists’ roles in advising athletes and making them not inadvertently consume prohibited substances. In my opinion, I am reluctant to say that because health behavior has components more than knowledge. They can understand correctly but still misuse it because of their attitude. More specifically, your research surveys only community pharmacists and does not include any athletes. So, the study results cannot infer the behavior of athletes.

Methodology

3. In the data collection subsection. The required characteristics of participants, i.e., registered pharmacists and working in the community pharmacy for at least one year, might be more appropriate to present in the study design and sampling subsection as eligibility criteria after the sampling method.

4. In the data collection subsection. You already answered another reviewer about the respondent explanatory statement about their honesty in answering the questionnaire. Suppose you include this detail in this section. In that case, it will help readers to understand the circumstance when respondents answer the questionnaire.

5. Why do you calculate the sample size? What is the rationale or parameters you include in this calculation? It might be easier to understand if you describe the purpose of the calculation. For example, to test a hypothesis, etc.

6. I think some of the demographic characters, i.e., the experience of practicing abroad and professional membership, are irrelevant to the doping knowledge. You should explain the rationale to include these characters in the introduction. Otherwise, it might be significant by chance alone without real association.

7. I puzzle about the way you scored this questionnaire’s answers. Is “not sure the answer” not equivalent to “the wrong answer”? Also, the cut point. Is this the standard way of using this questionnaire?

8. Why do you determine the association between the score and the demographic characteristics? It is not present in your study objective. Also, some characters look irrelevant, as I mention above. I saw you discuss these associations in the discussion section, but it does not make sense to me. Professional organizations in your study do not have a purpose for anti-doping. If it is an anti-doping organization, this might explain why?

Results

9. You have a pilot study. So, you have to mention it in the methodology section.

10. According to the response rate, we already know that the way respondents respond to the questionnaire affects the generalizability of the results, and you use both hardcopy and online forms. So, you should report the overall response rate and the response rate of hardcopy and online.

11. Subsection the respondents’ understanding of the term ‘doping.’ In your study. You use the question, “Have you heard about the term doping?”. You cannot conclude that the respondent who says yes understands what doping is. Heard is not equivalent to understanding, am I right? So, it would be best if you changed understanding to familiar or anything similar.

12. I found many typos or discordant results. For example, the percentage of respondents who knew about anabolic-androgenic steroids in the text (98.5%) is inconsistent with the table (95.8%). As well as your box-plot, IQR in the picture is approximately 17, but IQR in the table is 6. You should double-check your results again.

13. What is the meaning of U and H from statistic tests? If you can interpret it, you should do so. But if it is only raw statistical results like degree of freedom, you can omit it.

Discussion

14. I think reference number 26 should move to the end of the sentence, “Anabolic-androgenic steroids, diuretics, and stimulants were the three most commonly detected classes of drugs in doping tests according to WADA.” Rather than, “This probably explains why these drugs were readily identified by most of the respondents as prohibited substances in sports” because the latter sentence is your speculation.

15. Some of your claims are not supported by the results. For example, this explains why nicotine, NSAID, caffeine, and alcohol were sometimes mistaken as prohibited substances, considering their negative or enhancing effects on athletes’ sports performance. We do not know that because the athletes do not participate in this study. So, you cannot explain by using the results. It is only speculation.

16. The discussion part can be more concise. The way it is right now is like the key answer for grading the exam. I think some of the information in the discussion part is irrelevant to this study. For example, you state that inhaled glucocorticoids are underused in treating asthma in athletes. What is the connection? Does athlete think the inhaled glucocorticoids are prohibited, making them underused? However, it has nothing to do with this study.

17. You should discuss the anti-doping content in your bachelor’s degree curriculum. This might shed light on the readers as the anti-doping content is not required in many countries for a PharmD degree.

18. Limitation comes before the conclusion.

Reference

19. Lastly, you should cite each of them in the same format.

Sincerely,

7. PLOS authors have the option to publish the peer review history of their article (what does this mean?). If published, this will include your full peer review and any attached files.

Reviewer #3: No

Reviewer #4: **Yes: **Dan Tran, PharmD

Reviewer #5: No

---

## [Author Response · Author response to Decision Letter 1]

20 Apr 2022

Response to Reviewers 

Reviewer #3: 

The manuscript describes the doping problem at hand in Malaysia and the reason why community pharmacist engagement is crucial in combating the problem. The reviewer queries have been adequately addressed. The only fault I found was with a sentence in the abstract which needs correction. Please fix this at line 9 of the abstract. "With median years of practising as a community pharmacist for six years". The "for" in this phrase must be replaced with "of".

Author’s feedback: 

The phrase was replaced as suggested (page 3 line 52).

Reviewer #4: 

1. The introduction was significantly improved. However, I believe the background, significance, and rationale of the study can be even more improved. I would suggest that the authors discuss specifically medications that are often associated with inadvertent doping to give some context. This should be categorized into over-the-county versus prescription drugs.

Author’s feedback:

Some examples of the medications that are associated with inadvertent doping was added (page 4 line 91-94).

2. I also believe that a clear description of the roles of community pharmacists should be clearly mentioned, potentially in a section under Setting to provide some contextual understanding about how medications are dispensed (with or without a prescription), etc. In essence, response to Review #1, question 5, should be inserted into the text.

Author’s feedback:

In the response to Review #1, question 5, we explained on the categories of medications that are dispensed by the community pharmacists which are Group A, B, C, D, and non-poison medications (as listed below). However, this information is not included in the manuscript as it not directly relevant to the study findings. We have also pointed out that the study involved pharmacists working in the community setting (Lines 144-146).

Malaysia Regulatory Classification

Regulatory Classification for Malaysia

A - Group A

Can only be sold by a licensed wholesaler to a pharmacist or to another licensed wholesaler or by a licensed wholesaler to be immediately exported to a purchaser outside Malaysia.

B - Group B

Can be dispensed only against prescription of a Registered Medical Practitioner, Dentist, or Veterinary Surgeon, as the case may be and with the prescription in the correct form as required by the law.

C - Group C

Can only be sold as a dispensed medicine with entry in the Prescription Book.

D - Group D

Can only be sold as a dispensed medicine with an entry in the Poisons Book.

P2 - Part II Poisons

Retail sale restricted to Poison Licence Holder. Labelling requirements only.

NP- Non-scheduled Poisons

Non-scheduled poisons or over the counter products for retail sale.

3. It is still quite unclear to me how community pharmacists will be directly involved to eliminate inadvertent doping based on the data presented. While you have pointed out that more education can be provided, what practical tasks are you recommending that pharmacists do? For example, if someone comes into the pharmacy, are there standardized questionnaire to identify that (1) this is an athlete, and (2) they may be using drugs that lead to intentional or inadvertent doping. What interventions exist in the community pharmacy literature that have been shown to have impact? This is still very unclear to me... If you are to keep the study title as is: "Engaging Community Pharmacists to Eliminate Inadvertent Doping in Sports: A Study of their Knowledge on Doping", the "engaging" component will need additional justification and literature support. 

Author’s feedback:

We have extended the introduction to explain how the community pharmacists may take part in preventing inadvertent doping (page 4 line 87-91). We have also added a short paragraph at the discussion to highlight on the potential roles of pharmacist in prevention of doping in sports (page 33 line 599-608).

Reviewer #5: 

Introduction

1. The rationale of this study. You already mentioned that Chiang et al. had surveyed on this particular subject. What is the difference between the previous survey and yours? If you emphasize this point, it will make the gap or rationale of your study much more transparent.

Author’s feedback: 

In the current study, we wrote a more comprehensive set of questions for measuring the knowledge level. Chiang et al. assessed respondents’ knowledge of prohibited substances, WADA, NADO, and TUEs in Kuala Lumpur leaving out the roles of WADA, doping violations, ABP, and the doping situation in Malaysia. We had also expanded the research to all the other states in Malaysia (page 6 line 119-121).

2. You state that your study highlights community pharmacists’ roles in advising athletes and making them not inadvertently consume prohibited substances. In my opinion, I am reluctant to say that because health behavior has components more than knowledge. They can understand correctly but still misuse it because of their attitude. More specifically, your research surveys only community pharmacists and does not include any athletes. So, the study results cannot infer the behavior of athletes.

Author’s feedback: 

The aim of the study is to measure the level of knowledge and readiness of community pharmacists to engage with athletes. We take note of the suggestion regarding health-related behaviour and the appropriate preventive measures to curb drug abuse among athletes can be investigated in another study.

Methodology

3. In the data collection subsection. The required characteristics of participants, i.e., registered pharmacists and working in the community pharmacy for at least one year, might be more appropriate to present in the study design and sampling subsection as eligibility criteria after the sampling method.

Author’s feedback: 

We have moved the characteristics of the participants as suggested to study design and sampling (page 8 line 144-146).

4. In the data collection subsection. You already answered another reviewer about the respondent explanatory statement about their honesty in answering the questionnaire. Suppose you include this detail in this section. In that case, it will help readers to understand the circumstance when respondents answer the questionnaire.

Author’s feedback:

We have added the explanatory statement in the “Methods” section as suggested (page 8 line 157-159). This was also mentioned in the “Limitation” section (page 33 line 614-620).

5. Why do you calculate the sample size? What is the rationale or parameters you include in this calculation? It might be easier to understand if you describe the purpose of the calculation. For example, to test a hypothesis, etc.

Author’s feedback: 

Calculating the sample size is to ensure we recruited sufficient numbers of participants to generalize the study findings to the whole population (page 8 and 9 line 167-168).

6. I think some of the demographic characters, i.e., the experience of practicing abroad and professional membership, are irrelevant to the doping knowledge. You should explain the rationale to include these characters in the introduction. Otherwise, it might be significant by chance alone without real association.

Author’s feedback: 

We explained the rationale to include some of the demographic information as suggested (page 10 line 179-181). We proposed that experience of practising abroad can provide the pharmacists opportunity to be exposed to different experiences with athletes. Meanwhile, pharmacist in Malaysia would need to fulfil the Continuous Professional Development (CPD) credit points in order to renew the license. Thus, we propose that pharmacists with professional membership would receive more information on the courses provided by the society and thus may have better knowledge in various topics. The association of these demographic characteristics with knowledge levels were presented in Table 4.

7. I puzzle about the way you scored this questionnaire’s answers. Is “not sure the answer” not equivalent to “the wrong answer”? Also, the cut point. Is this the standard way of using this questionnaire?

Author’s feedback: 

The reason of awarding 1 mark for "not sure" is we think the participants had a certain degree of knowledge on the topic but is "unsure" of the answer. Even if the participants answered "not sure" for all the questions, they would only score 34 (which is classified as poor level as well). The cutting point is based on previous literatures as cited (page 10 line 199-204). 

8. Why do you determine the association between the score and the demographic characteristics? It is not present in your study objective. Also, some characters look irrelevant, as I mention above. I saw you discuss these associations in the discussion section, but it does not make sense to me. Professional organizations in your study do not have a purpose for anti-doping. If it is an anti-doping organization, this might explain why?

Author’s feedback: 

Our hypothesis is that members of professional organisations may in general be more motivated to learn new things to increase their knowledge because of requirements for continuous professional development. Please also refer to our response to comment no 6.

Results

9. You have a pilot study. So, you have to mention it in the methodology section.

Author’s feedback:

We have added on the explanation on the pilot study as suggested (page 8 line 151-154).

10. According to the response rate, we already know that the way respondents respond to the questionnaire affects the generalizability of the results, and you use both hardcopy and online forms. So, you should report the overall response rate and the response rate of hardcopy and online.

Author’s feedback:

We distributed the questionnaire through a variety of social media platforms and WhatsApp groups. We used the snowballing and convenience sampling methods to distribute online and physical copies of the questionnaire. This makes it impossible to accurately calculate the response rate.

11. Subsection the respondents’ understanding of the term ‘doping.’ In your study. You use the question, “Have you heard about the term doping?”. You cannot conclude that the respondent who says yes understands what doping is. Heard is not equivalent to understanding, am I right? So, it would be best if you changed understanding to familiar or anything similar.

Author’s feedback: 

We have reworded ‘understanding’ to ‘familiarity’ as suggested (page 13 line 248).

12. I found many typos or discordant results. For example, the percentage of respondents who knew about anabolic-androgenic steroids in the text (98.5%) is inconsistent with the table (95.8%). As well as your box-plot, IQR in the picture is approximately 17, but IQR in the table is 6. You should double-check your results again.

Author’s feedback 

We have checked the manuscript and corrected the inconsistencies. The percentage of respondents who knew about AAS is corrected to 95.8% (page 15 line 267). Figure 1 is removed. 

13. What is the meaning of U and H from statistic tests? If you can interpret it, you should do so. But if it is only raw statistical results like degree of freedom, you can omit it.

Author’s feedback: 

U is the symbol used in Mann-Whitney U test while H is the symbol used in Kruskal-Wallis test; legends were inserted at the bottom of the table 5.

Discussion

14. I think reference number 26 should move to the end of the sentence, “Anabolic-androgenic steroids, diuretics, and stimulants were the three most commonly detected classes of drugs in doping tests according to WADA.” Rather than, “This probably explains why these drugs were readily identified by most of the respondents as prohibited substances in sports” because the latter sentence is your speculation.

Author’s feedback:

We have moved the reference as suggested and rewrite some part of the discussion to make it more concise and relevant to the study findings.

15. Some of your claims are not supported by the results. For example, this explains why nicotine, NSAID, caffeine, and alcohol were sometimes mistaken as prohibited substances, considering their negative or enhancing effects on athletes’ sports performance. We do not know that because the athletes do not participate in this study. So, you cannot explain by using the results. It is only speculation.

Author’s feedback: 

We have removed this paragraph and refine the discussion. In the original version of the discussion, what we intended to point out was that those substances were sometimes mistaken by pharmacists (not athletes) as being prohibited in sports. So, whether athletes took part in the study is irrelevant.

16. The discussion part can be more concise. The way it is right now is like the key answer for grading the exam. I think some of the information in the discussion part is irrelevant to this study. For example, you state that inhaled glucocorticoids are underused in treating asthma in athletes. What is the connection? Does athlete think the inhaled glucocorticoids are prohibited, making them underused? However, it has nothing to do with this study.

Author’s feedback: 

We have removed this paragraph and refined the discussion.

17. You should discuss the anti-doping content in your bachelor’s degree curriculum. This might shed light on the readers as the anti-doping content is not required in many countries for a PharmD degree.

Author’s feedback: 

We have added a short explanation on the anti-doping course as offered by the university in Malaysia. The course is commonly known as “Drugs in Sports” in most of the university (page 32 line 573-576). 

18. Limitation comes before the conclusion.

Author’s feedback:

We have moved the limitation as suggested.

Reference

19. Lastly, you should cite each of them in the same format.

Author’s feedback: 

All the references were cited as Vancouver format and any new references added were cited in the desired format.

---

## [Decision Letter · Decision Letter 2]

11 May 2022

Engaging Community Pharmacists to Eliminate Inadvertent Doping in Sports: A Study of their Knowledge on Doping

PONE-D-21-01079R2

Dear,

We’re pleased to inform you that your manuscript has been judged scientifically suitable for publication and will be formally accepted for publication once it meets all outstanding technical requirements.

Kind regards,

Muhammad Shahzad Aslam, Ph.D.,M.Phil., Pharm-D

Academic Editor

PLOS ONE

Additional Editor Comments (optional):

Reviewers' comments:

Reviewer's Responses to Questions

**Comments to the Author**

1. If the authors have adequately addressed your comments raised in a previous round of review and you feel that this manuscript is now acceptable for publication, you may indicate that here to bypass the “Comments to the Author” section, enter your conflict of interest statement in the “Confidential to Editor” section, and submit your "Accept" recommendation.

Reviewer #5: All comments have been addressed

2. Is the manuscript technically sound, and do the data support the conclusions?

Reviewer #5: Yes

3. Has the statistical analysis been performed appropriately and rigorously? 

Reviewer #5: Yes

4. Have the authors made all data underlying the findings in their manuscript fully available?

Reviewer #5: Yes

5. Is the manuscript presented in an intelligible fashion and written in standard English?

Reviewer #5: Yes

6. Review Comments to the Author

Reviewer #5: I really appreciate your modifications and explanations. This version is easier to understand. I think your work can inspire community pharmacists around the world to expand their roles.

7. PLOS authors have the option to publish the peer review history of their article (what does this mean?). If published, this will include your full peer review and any attached files.

Reviewer #5: No

---

## [Editor Report · Acceptance letter]

2 Jun 2022

PONE-D-21-01079R2 

Engaging Community Pharmacists to Eliminate Inadvertent Doping in Sports: A Study of their Knowledge on Doping 

Dear Dr. Mhd Ali:

I'm pleased to inform you that your manuscript has been deemed suitable for publication in PLOS ONE. Congratulations! Your manuscript is now with our production department. 

Kind regards, 

on behalf of

Dr. Muhammad Shahzad Aslam 

Academic Editor

PLOS ONE